# Comparative Proteomic and Metabolomic Analysis of Human Osteoblasts, Differentiated from Dental Pulp Stem Cells, Hinted Crucial Signaling Pathways Promoting Osteogenesis

**DOI:** 10.3390/ijms22157908

**Published:** 2021-07-24

**Authors:** Slavomíra Nováková, Maksym Danchenko, Terézia Okajčeková, Eva Baranovičová, Andrej Kováč, Marián Grendár, Gábor Beke, Janka Pálešová, Ján Strnádel, Mária Janíčková, Erika Halašová, Henrieta Škovierová

**Affiliations:** 1Biomedical Centre Martin, Jessenius Faculty of Medicine in Martin, Comenius University in Bratislava (JFM CU), Malá Hora 4C, 036 01 Martin, Slovakia; tereza.okj@gmail.com (T.O.); eva.baranovicova@uniba.sk (E.B.); marian.grendar@uniba.sk (M.G.); sopkova24@uniba.sk (J.P.); jan.strnadel@uniba.sk (J.S.); erika.halasova@uniba.sk (E.H.); 2Plant Science and Biodiversity Center, Slovak Academy of Sciences, Dúbravská cesta 9, 845 23 Bratislava, Slovakia; maksym.danchenko@savba.sk; 3Institute of Neuroimmunology, Slovak Academy of Sciences, Dúbravská cesta 9, 845 10 Bratislava, Slovakia; andrej.kovac@savba.sk; 4Institute of Molecular Biology, Slovak Academy of Sciences, Dúbravská cesta 21, 845 51 Bratislava, Slovakia; gabor.beke@savba.sk; 5Department of Stomatology and Maxillofacial Surgery, University Hospital in Martin and JFM CU, Kollárova 2, 036 01 Martin, Slovakia; maria.janickova@uniba.sk; 6Department of Medical Biology, Jessenius Faculty of Medicine in Martin, Comenius University in Bratislava (JFM CU), Malá Hora 4C, 036 01 Martin, Slovakia

**Keywords:** DPSCs, osteogenic differentiation, label-free quantification, NMR, bone metabolism, mineralization phase, late osteoblast

## Abstract

Population aging has been a global trend for the last decades, which increases the pressure to develop new cell-based or drug-based therapies, including those that may cure bone diseases. To understand molecular processes that underlie bone development and turnover, we followed osteogenic differentiation of human dental pulp stem cells (DPSCs) using a specific induction medium. The differentiation process imitating in vivo osteogenesis is triggered by various signaling pathways and is associated with massive proteome and metabolome changes. Proteome was profiled by ultrahigh-performance liquid chromatography and comprehensively quantified by ion mobility-enhanced mass spectrometry. From 2667 reproducibly quantified and identified proteins, 432 were differentially abundant by strict statistic criteria. Metabolome profiling was carried out by nuclear magnetic resonance. From 27 detected metabolites, 8 were differentially accumulated. KEGG and MetaboAnalyst hinted metabolic pathways that may be involved in the osteogenic process. Enrichment analysis of differentially abundant proteins highlighted PPAR, FoxO, JAK-STAT, IL-17 signaling pathways, biosynthesis of thyroid hormones and steroids, mineral absorption, and fatty acid metabolism as processes with prominent impact on osteoinduction. In parallel, metabolomic data showed that aminoacyl-tRNA biosynthesis, as well as specific amino acids, likely promote osteodifferentiation. Targeted immunoassays validated and complemented omic results. Our data underlined the complexity of the osteogenic mechanism. Finally, we proposed promising targets for future validation in patient samples, a step toward the treatment of bone defects.

## 1. Introduction

The bone is a highly specific tissue that plays an essential role in the human body. It consists of specialized cells (osteoblasts, osteocytes, and osteoclasts), a mineralized and non-mineralized matrix, spaces including the vascular canals, bone marrow cavity, ducts, and gaps [1]. Bones form a skeleton that provides protection, mechanical support for soft tissues, and allows mobility. In addition, it is a mineral reservoir that supports calcium and phosphate homeostasis; therefore, the skeleton is closely linked to metabolic homeostasis as well as to the dynamic regulation by hormones and nutrients [2]. The main factor that affects the amount of bone mass is genetics. However, other variable factors such as diet, environmental involvement, exercises, current diseases, and drugs could influence bone mass [3]. Skeleton is an active organ: during growth and development, two bone-formation processes (modeling and remodeling) are involved in its sculpturing. Modeling [1] takes place during childhood and adolescence [2]. During this process, the bone mass is removed from one location and deposited to another location within the same bone for creating proper shape and size. After maturation, bone regeneration relies on a dynamic balance between bone formation (ossification) and resorption, mediated by osteoblasts and osteoclasts, respectively. This process is called remodeling [4]. The imbalance in such regulation leads to many bone diseases; for example, osteopetrosis due to excessive bone formation [5] or, in contrast, postmenopausal osteoporosis due to massive bone resorption [6].

One of the major recent challenges of regenerative medicine is bone regeneration/repair after various pathological problems such as fracture, metabolic diseases, osteoporosis, or tumors. In clinical practice, procedures for the treatment of these defects include drug application [7,8,9], using bone grafts, or the application of stem cells [10,11]. The use of medications for treating osteoporosis and fractures is limited by their efficacy and variable toxicities [3,12,13]. Therefore, clinicians recommend focusing on the prevention of fractures and defects by adopting a healthy lifestyle, using adequate nutritional supplements (calcium and vitamin D), avoiding smoking and alcohol consumption, and controlling weight [14].

The modern alternative to drug applications is tissue engineering using stem cells. Adult stem cells (in particular mesenchymal stem cells, MSCs) are very popular due to their self-renewal, differentiation capacity, multipotentiality, immune-modulatory and anti-inflammatory properties, and lack of ethical restrictions [15]. The MSCs can be directly injected into the injured bone, systemically infused, or loaded on three-dimensional scaffolds before implantation [16]. The latest option is preferable because it supports the growth of various cell types and this complex will trigger the healing capacity of bone tissue [17,18]. Human dental pulp stem cells (DPSCs), oral dental-derived stem cells, are part of the dental pulp, which ensures the formation, nutrition, protection, and restoration of teeth. Dental pulp produces dentin and maintains its physiological and biological vitality [19]. As DPSCs have the properties of MSCs, aside from their odontogenic potential, they can differentiate in vitro into osteoblasts, chondrocytes, myocytes, cardiomyocytes, active neurons, and hepatocyte-like cells, thus promoting healing of tissues [20].

To develop and practically implement the new advanced cell-based or drug-based strategies, it is necessary to understand molecular mechanisms responsible for natural osteogenesis and bone regeneration. Bone healing is a complex process including inflammation, repair, and remodeling. Various cell types (in particular MSCs) and multiple extracellular and intracellular signaling networks are involved in this well-organized physiological process [16,17]. Osteogenic microenvironment is often imitated in vitro by specific induction media [21]. The exact molecular mechanism by which cells differentiate and form a mineralized extracellular matrix is not yet fully understood.

Of note, the differences during in vitro differentiation of DPSCs into osteoblasts or odontoblasts are minor; consequently, inducing media are similar. While some researchers modulated in vitro odontogenesis by β-glycerophosphate [22], others used the medium with this supplement as osteogenic [23,24]. Specific successful differentiation of DPSCs with the proper structural arrangement into osteoblasts/odontoblasts is ensured only in vivo [25]. In the recent review, the authors argued that bone and dentin share similar characteristics in their composition, mineralization, biomarkers, and clinical manifestation of mineralization defects. It is difficult to discriminate bone/tooth differentiation markers because these proteins are expressed in both tissues [26]. Analogously, signaling pathways involved in osteogenic/odontogenic differentiation overlap [17,27,28]. Moreover, a large share of information about dentin mineralization was obtained from in vitro and in vivo bone studies [29]. Therefore, DPSCs are applicable not only in cranio-maxillofacial hard-tissue repair [30,31] but also have the potential for the treatment of various bone defects [32]. For example, dentin was applied as a graft to repair calvarial bone defect [33]. In this study, we focused on in vitro differentiation of DPSCs from the osteogenic perspective.

Omics technologies give the possibility of a complex view of the biological issue. Herein, we analyzed proteome and metabolome changes in DPSCs during osteogenic differentiation to understand the molecular background of the bone regeneration process. Comprehensive data were obtained using label-free mass spectrometry and NMR quantification. Next, we validated findings by the detection of specific proteins using Western blot analysis and immunocytochemistry. A complex understanding of signaling pathways that regulate the osteodifferentiation of stem cells advanced knowledge of the molecular mechanisms of bone development as well as offered ideas for more effective therapies for bone diseases.

## 2. Results and Discussion

The entire experimental workflow is shown in Figure 1 to illustrate the specific design of this work.

Bone remodeling is a complex process controlled by multiple local and systemic factors. In order to study the mechanisms that regulate osteodifferentiation, DPSCs were grown in OsteoMAX-XF differentiation medium selected in our previous study [21]. The efficiency of differentiation was demonstrated (i) microscopically, (ii) spectrophotometrically, and (iii) immunocytochemically.

### 2.1. DPSCs Possess In Vitro Osteogenic Potential

The differentiation process was monitored for 24 days based on cell morphology. Data in the literature point to cellular morphology correlations with the osteogenic differentiation potential as well as biochemical markers and provides strong predictive evidence in regenerative medicine [34]. During our experiment, cells showed the phenotype which was typical for a particular day of osteogenic development. This pattern was reproducible, as shown by three independent biological replicates (Figure 2 and Appendix A).

Osteogenic differentiation consists of several phases. Our work focused on the 24th day after the specific medium application. It falls into the mineralization phase, which is characterized by the production of osteocalcin (OSTCN) and osteopontin secreted by late osteoblasts as well as calcium and phosphate deposition [35]. On day 24, to confirm the proper differentiation process before performing omic analysis, OSTCN, bone sialoprotein 2 (BSP II), and bone morphogenetic protein 2 (BMP-2) as typical osteogenic markers [36] were imaged by immunocytochemistry. All three proteins showed high accumulation in osteodifferentiated cells compared to the weak signal in control (Figure 3).

Additionally, we followed the dynamics of calcium nodule production during the experimental time course. We revealed a remarkable production of mineral deposits of more than 100 times higher in differentiated cells compared to DPSCs on the 11th day after application of the osteogenic medium. Accumulation of deposits further increased almost three times from the 11th to the 24th day (from 139:1 to 396:1) (Figure 4).

### 2.2. Overview of Quantitative Proteomic and Metabolomic Profiling of DPSCs upon In Vitro Osteogenic Differentiation

To analyze the mechanisms of the differentiation process, a label-free quantitative approach was employed to acquire the peptide and fragment mass spectra. Dedicated software Progenesis QI with advanced chromatographic alignment, missing values free peak detection, and relative protein abundance determination, relying on the integration of peak areas of unique peptides, was used to process the mass spectrometry data. We identified and quantified 2667 proteins in the control and osteodifferentiated samples (Appendix A). Among them, 432 proteins were differentially regulated based on criteria of statistical significance (*p* ≤ 0.01) and at least 1-fold of log2 transformed ratio (Appendix A). Principal component analysis (PCA) revealed proper clustering of experimental replicates in two-dimensional space, increasing confidence in findings (Appendix AA). Then, proteins that met these stringent filtering criteria were categorized: (i) based on a biological process and (ii) based on signaling pathway. Proteins were clustered into twelve functional categories using GOTermFinder. The biological process categories affected during osteogenesis included cellular process (GO:0009987, 392 proteins), localization (GO:0051179, 223 proteins), response to stimulus (GO:0050896, 196 proteins), biological regulation (GO:0065007, 194 proteins), development (GO:0032502, 190 proteins), transport (GO:0006810, 182 proteins), cellular component organization (GO:0071840, 182 proteins), metabolism (GO:0008152, 166 proteins), cell death (GO:0008219, 89 proteins), secretion (GO:0046903, 77 proteins), immune system process (GO:0002376, 42 proteins), and cellular respiration (GO:0045333, 19 proteins). Fine granularities of ontologies are presented in Appendix A. To enhance the biological context, differentially abundant proteins were also grouped into 265 metabolic and signaling pathways using KEGG (Appendix A). The identified molecular pathways after filtering included several potentially crucial cascades promoting osteogenic differentiation. We described them in detail in the following section.

In parallel, we quantified 27 metabolites in the control and osteodifferentiated samples (Appendix A). Among them, only eight were differentially regulated based on criteria of statistical significance (adjusted *p* ≤ 0.05) and at least 1-fold of log2 transformed ratio (Appendix A). Then, metabolites satisfying these stringent filtering criteria were categorized into 14 metabolic pathways using MetaboAnalyst (Appendix A). As in the proteins case, the PCA revealed proper clustering of three replicates in two-dimensional space (Appendix AB).

### 2.3. Major Signaling Pathways Involved in Osteogenesis Were Well Represented among Differentially Abundant Proteins

Initially, we focused on identifying and selecting several known signaling pathways highlighted in the literature (Table 1). We found several of the major pathways that were already described in connection with osteodifferentiation [17,37]. Namely, Ca^2+^, Wnt, TGF-β, PI3K-Akt, JAK-STAT, and MAPK signaling pathways were identified as affected in the mineralization phase after osteogenic medium induction on the 24th day (Figure 5, Table 1). Apart from the PI3K-Akt pathway (with a roughly equal number of proteins accumulated in control and osteodifferentiated cells), JAK-STAT and MAPK signaling cascades were both downregulated in differentiated cells. The remaining pathways were upregulated in differentiated cells, confirming their pivotal roles in osteogenesis. However, these pathways were not significantly enriched apart from JAK-STAT. Of note, the cascades crosstalk, hence form a signaling network. Differentially abundant proteins from these pathways may become either diagnostic markers in patients’ samples or even therapeutic targets.

*JAK-STAT signaling pathway*. MAPK and JAK-STAT signaling cascades were apparently downregulated in osteodifferentiated cells. JAK-STAT signaling is involved in many pathologies as well as bone homeostasis. STAT family consists of several members, particularly STAT1 and STAT3, that variously influence bone development [38]. Interestingly, it was shown that STAT1 absence led to the acceleration of bone healing [39]. Furthermore, STAT1/3 inhibition increased osteoblast activity, associated with OSTCN and alkaline phosphatase production [40]. STAT3 also controls osteoclast differentiation, bone degeneration, and homeostasis [41]. Our data are in line with those reports. We identified four members of JAK-STAT pathway (Table 1), signal transducer and activator of transcription 1 (STAT1), signal transducer and activator of transcription 3 (STAT3), four and half LIM domains 1 (FHL1), and glial fibrillary acidic protein (GFAP) as lower abundant in differentiated cells (−1.55, −1.11, −1.65, and −1.08-fold decrease, respectively). We hypothesize that in the mineralization phase, downregulation of STAT1 and STAT3 positively affects osteogenesis.

*Ca^2+^ signaling pathway*. Calcium and phosphate, major units of mineral metabolism, are closely linked to intermediary metabolism. Intracellular calcium also works as a secondary messenger that mediates the effect of membrane signals on the secretion of several mediators (e.g., insulin and epinephrine). Circulating calcium levels are tightly controlled through the mineral hormones PTH, 1,25-dihydroxyvitamin D, and calcitonin [42]. A higher level of cytoplasmic Ca^2+^ promotes phospholipase C activity and inositol-1,4,5-trisphosphate (IP3) signaling. This signaling leads to a release of Ca^2+^ from intracellular stores (mainly from the endoplasmic reticulum). Then, increased cytoplasmic Ca^2+^ further promotes regulation of several factors, such as IGF1 and IGF2, VEGF, TGF-β, BMP-2, and BMP-4, involved in other osteogenic signaling pathways [17]. We identified nine members of Ca^2+^ signaling pathway (Table 1). Specifically, voltage-dependent anion channels 1, 2, 3 (VDAC1, VDAC2, VDAC3; 1.18, 1.14, and 1.36-fold increase, respectively), solute carrier family 25 members 4 and 6 (SLC25-4, SLC25-6 both 1.05-fold increase), IP3 receptor type 2 (4.92-fold increase), aspartate beta-hydroxylase (ASPH, 1.07-fold increase), and G protein alpha subunits 11 and 14 (GNA11 and GNA14, −3.15 and −1.99-fold decrease, respectively). The identification of VDACs, ion channels located in the outer mitochondrial membrane, and SLC25s, solute carriers responsible for mitochondrial ATP transport, points to the role of mitochondria in the Ca^2+^ influx and efflux as more important than the endoplasmic reticulum during osteodifferentiation. This interpretation corresponds with the conclusions of Alves et al. [43]. Our results were strengthened by determining the IP3 receptor (validation of proteomic data) and BMP-2 (complementation) by Western blotting. Densitometry showed a statistically significant 4.32 and 7.65-fold increase, both *p* = 0.044 in osteodifferentiated cells compared to controls (Figure 6). A better understanding of the Ca^2+^-sensing mechanism may lead to novel targets for promoting osteogenesis because of close links with other cascades. For instance, it was reported that increased Ca^2+^ through its channels enhanced bone formation and prevented estrogen deficiency-induced bone loss [44].

*Wnt signaling pathway*. This pathway plays an important role during the growth, development, and homeostasis of numerous organs, including the bone system [45]. Wnts belong to a family of ligands for membrane-spanning frizzled (Fzd) receptors. Wnts can activate at least three signaling subcascades: Wnt/Ca^2+^ (noncanonical), Wnt/β-catenin (canonical), and Wnt/planar polarity (noncanonical) [17]. We detected a single Wnt protein (Table 1), Wnt family member 5A (1.24-fold increase in osteoblasts). Considering upregulated Ca^2+^ signaling, we suggest activation of noncanonical subcascade. It is initiated by Wnt ligands binding to Fzd receptors followed by the activation of G protein and downstream events releasing intracellular calcium from the endoplasmic reticulum [46]. Of note, it was reported that Wnt5A-induced calcium-dependent pathway promoted osteoblast differentiation through the upregulation of Lrp5 and Lrp6 and enhanced Wnt/β-catenin signaling [47]. Additionally, other studies pointed to the regulation of the Wnt signaling during osteogenesis; thus, it became an attractive therapeutic target for the treatment of bone defects [46].

*TGF-β cascade*. BMP/TGF-β signaling is a crucial network associated with maintaining bone development by regulating the balance between bone resorption and bone formation. Signaling takes place through the canonical (Smad-dependent) or noncanonical (Smad-independent) pathway. BMPs and TGF-βs are cytokines belonging to TGF-β superfamily [48]. It includes several subgroups, among others activins and inhibins. We identified only two members of TGF-β pathway (Table 1), decorin and inhibin subunit beta A, both accumulated in osteodifferentiated cells (1.75 and 5.20-fold increase, respectively). There is only scarce information in the literature about the modulation of osteogenesis by activins and inhibins, such as they act as modulators of TGF-β and BMP. Particularly, Gaddy-Kurten et al. reported that inhibins A and β_A_ repress osteoblast differentiation and reduce alkaline phosphatase activity [49]. Decorin is a small leucine-rich proteoglycan associated with extracellular matrix deposition in the mineralization stage of bone and dentin development. It preferentially binds hydroxyapatite, regulating crystal size and morphology [50]. Covalent binding of decorin to TGF-β resulted in increased bone mass associated with a decreased level of TGF-β. Active TGF-β promotes osteoclast differentiation at a low dose but inhibits this process at a high dose [51]. Thus, we argue that a higher abundance of inhibin and decorin indicates the termination of active differentiation of osteoblasts, which reached the mineralization phase. Additionally, decorin is a potential marker or target to improve mineralization in bones. For instance, the authors showed low expression of decorin in the skin of patients with osteoporosis imperfecta [52]. A study on pigs demonstrated the importance of dentin sialophosphoprotein (DSPP) in maintaining TGF-β1 activity in dentin [53]. We detected (by complementary Western blotting) accumulation of its cleavage product (likely dentin sialoprotein, DSP) at 37 kDa in osteodifferentiated cells (14.91-fold increase, *p* = 0.0002) (Figure 6). The size of the observed DSP band is the same as reported by Yamakoshi et al. [54] as well as in DPSCs study of Teti et al. [55]. Our interpretation is that accumulation of decorin led to inactivation of TGF-β1, thus higher abundance of DSP in osteoblasts. Although the DSPP (also DSP) protein is primarily associated with odontogenesis, it was identified also in bones with function in hard tissue development. Verdelis et al. demonstrated DSPP expression at low level in the long bone; moreover, its disruption in mice resulted in modest but significant negative changes [56]. Jani et al. suggested that DSPP and DMP1 may function synergistically. Specifically, they showed that *Dspp* rescued the long bone defects in *Dmp1*-deficient mice [57].

*MAPK signaling pathway*. Mitogen-activated protein kinase (MAPK) cascade shares a few members with non-canonical TGF-β signaling. Seven from ten identified members were less abundant in osteodifferentiated cells (Table 1). Only insulin (INS), insulin-like growth factor 2 (IGF-2), and CD14 protein accumulated in osteoblasts (2.64, 1.12, and 2.42-fold increase, respectively). INS and IGF-2 are hormonal signaling molecules that affect a broad spectrum of metabolic and developmental processes. They regulate signaling from extracellular space until target expression in the cell. Several studies reported their impact on osteoblasts and bone development through PI3K-Akt and MAPK pathways [58,59]. Lower abundant proteins in osteoblasts included MAP3K20, MAP4K4, microtubule-associated protein tau, protein phosphatase 5, calcium voltage-gated channel, CRK proto-oncogene, and G protein subunit alpha 12 (−1.32, −3.12, −3.43, −1.17, −1.31, −1.08, and −1.73-fold decrease, respectively). The role of the MAPK pathway in osteoblasts remains controversial, as some studies have reported its stimulatory effect while others reported an inhibitory one [60]. Interestingly, the authors reported a direct connection of dentin matrix protein (DMP-1) to MAPK signaling activation and maintenance of osteocytes’ phenotype [61]. In contrast, another work suggested the involvement of DMP-1 as a negative regulator of bone formation [62]. Our data are consistent with the latter report. Complementary Western blotting showed a lower abundance of DMP-1 (−5.54-fold decrease, *p* = 0.18) in differentiated cells (Figure 6).

*PI3K-Akt pathway*. Phosphatidylinositol 3-kinase (*PI3K*) and its downstream targets, serine/threonine kinases (AKT), play an essential role in the differentiation of progenitor cells into mature osteoblasts [63]. We revealed nine members of this pathway (Table 1): four accumulated (insulin, insulin-like growth factor 2, heat shock protein 90, and laminin subunit beta 1) and five less abundant (collagen types 1/4/6 alpha 1 chain, laminin subunit beta 2, and reelin) in differentiated cells. Of note, several collagen types were reported as factors that promote osteodifferentiation [64,65]. We believe that a lower abundance of collagens in mature osteoblasts, together with accumulation of Hsp 90 (1.99-fold increase), a factor that induces osteoclast-associated genes and enhances osteoclastogenesis, indicates increased bone turnover. On the other hand, undifferentiated DPSCs (control), synthesize and secrete massive amounts of collagen, particularly types 1 and 6 [66]. Therefore, the lower level of collagen isoforms in differentiated cells is not surprising. Accumulation of Col6A1 in controls was validated by Western blotting (6.33-fold increase, *p* = 0.0002) (Figure 6).

Overall, our data on major signaling pathways connected with osteodifferentiation highlighted the following conclusions: (i) in the mineralization phase, downregulation of STAT1 and STAT3 positively affects osteogenesis; (ii) the accumulation of VDACs and SLC25s points to the primary role of mitochondria in the Ca^2+^ influx/efflux during osteodifferentiation; (iii) Wnt5a is an attractive therapeutic target for the treatment of bone defects; (iv) accumulation of decorin inactivated TGF-β, leading to accumulation of DSP in osteoblasts; (v) DMP-1 is a negative regulator of bone formation via MAPK cascade; and (vi) lower abundance of collagens belonging to PI3K-Akt cascade in mature osteoblasts indicates increased bone turnover.

### 2.4. Selected Pathways Potentially Associated with Bone Metabolism

Despite confirmation of several signaling pathways and molecules involved in osteogenesis, much remains unknown. We identified proteins in several cascades, which showed high enrichment scores in our analysis (Table 1 and Appendix A, Figure 5). They are not considered major, since little is known about their connection to bone metabolism. Therefore, they need further exploration.

#### 2.4.1. Prevention of Oxidation Stress by FoxO Cascade, Hormone Synthesis, and Mineral Absorption Are Factors That May Affect the Mineralization Phase of Osteodifferentiation

*FoxO signaling pathway and pathway for the proteasome*. Forkhead box O (FOXO) family transcription factors have an essential function in numerous biological processes, including cellular proliferation, differentiation, and apoptosis [67]. We identified four members of FoxO pathway (Table 1), protein arginine methyltransferase (PRMT1), superoxide dismutase (SOD2), INS more abundant (2.31, 3.17, 2.64-fold increase, respectively) and STAT3 less abundant (−1.11-fold decrease) in differentiated cells. FoxO signaling deficiency usually leads to bone mass reduction. Moreover, FOXOs maintain the lifespan of mature osteoblasts through antioxidants [68]. During differentiation ROS levels frequently increase due to enhanced cellular metabolism [69]. FOXO promotes the expression of antioxidant enzymes, which maintain low ROS levels, thus protecting the cells from osteoclastogenesis [70]. In line with these reports, we detected the accumulation of enzymes, antioxidant SOD2, and regulatory PRMT1, in differentiated cells. The authors reported that the latter methylated FOXO1 at conserved Arg248 and Arg250. This methylation directly blocked Akt-mediated phosphorylation of FOXO, which leads to translocation to the cytoplasm and subsequent degradation via the ubiquitin-proteasome system. Silencing of PRMT1 usually enhances proteasomal degradation of FOXO1 [71,72]. Methylation blocks the exclusion of FOXO1 from the nucleus, and consequently activates the expression of target genes, particularly, antioxidant enzymes [73]. This link is additionally supported by the identification of pathway for the proteasome downregulated in differentiated cells (Table 1). In addition, a study on the animal model showed that resveratrol prevented osteoporosis by increasing *FOXO1* transcription [74].

*IL-17 pathway*. Interleukin 17 (IL-17), produced by Th17 cells, is a family of pro-inflammatory cytokines having a role in bone remodeling. They enhance MSCF and RANKL production, which leads to the formation of osteoclasts with bone-resorbing function. Furthermore, IL-17 induces chemokines with an impact on osteogenic differentiation [75]. We detected accumulation of six proteins from IL-17 pathway in differentiated cells (Table 1); particularly, chemokines (CXCL1, CXCL5, CXCL6 1.16, 2.04, 2.67-fold increase, respectively), matrix metallopeptidases (MMP1 and MMP3, 3.09 and 4.04-fold increase, respectively), and heat shock protein 90 (1.99-fold increase). Unique chemokines specifically affect bone remodeling, such as CXCL1 and CXCL5 function in osteoclastogenesis [76]. Additionally, MMPs are recognized osteoclastic resorption factors. The authors reported that MMPs inhibition can even block osteoclastogenesis [77,78]. Accumulation of proteins belonging to IL-17 pathway suggests the developmental switch on 24 days after triggering differentiation. Interestingly, it was reported that 1,25-dihydroxyvitamin D3 and thyroid hormones modulated MMP13 [79]—a link to a biosynthetic pathway described in the following section.

*Biosynthesis of thyroid hormones and steroids*. Among other factors, bone turnover is regulated by numerous hormones which maintain a fragile balance between bone formation and resorption. Hormones can act directly or indirectly as growth factors, receptor-binding molecules, and synthesis modulators. They include thyroid and parathyroid hormones, estrogens, growth hormones, and 1,25-dihydroxyvitamin D3. Thyroid-stimulating hormone (TSH) works via TSH receptors, which are activated by protein disulfide isomerase [80]. Additionally, TSH affects secondary messengers (for example, Ca^2+^ signaling cascade through IP3) in extrathyroidal tissues. It was reported that in embryonic stem cells, TSH stimulated osteoblastic differentiation via protein kinase C and noncanonical Wnt pathway. TSH is directly connected to bone homeostasis and development, for example, thyrotoxicosis in adults was accompanied by secondary osteoporosis and increased risk of fractures [81]. We detected five proteins involved in thyroid hormone synthesis pathway all upregulated in differentiated cells (Table 1). Namely, albumin, heat shock protein 70, inositol 1,4,5-trisphosphate receptor type 2, protein disulfide isomerase, and transthyretin (2.45, 1.76, 4.92, 1.50, and 2.51-fold increase, respectively). Interestingly, human transthyretin, TSH transporter, is involved in the bone mineralization process and its destabilization leads to changed crystal morphology [82]. All steroid hormones are synthesized from cholesterol. We identified three members of the steroid biosynthesis pathway accumulated in differentiated cells (Table 1): 7-dehydrocholesterol reductase (DHCR7), methylsterol monooxygenase 1, and NAD(P)-dependent steroid dehydrogenase-like (2.77, 2.38, 2.47-fold increase, respectively), which are involved in cholesterol metabolism. In general, LDL cholesterol promotes osteoclastogenesis and vice versa, HDL cholesterol protects osteoblasts from apoptosis [83]. Interestingly, cholesterol-mediated degradation of DHCR7 switched the cholesterol synthesis to vitamin D production [84] causing improved bone formation [85]. DHCR7 is the terminal enzyme of cholesterol synthesis converting 7-dehydrocholesterol (7DHC) to cholesterol. Since our data showed a high abundance of DHCR7 and apolipoprotein A2 (ApoA2) from upregulated PPAR pathway (discussed below in Section 2.4.2) in osteoblasts, we hypothesize that the former routes ApoA2 to HDL cholesterol for stabilizing structure.

*Mineral absorption pathway*. Bone remodeling is an active energy-consuming process, which requires a supply of macro- and micronutrients such as vitamins and minerals. It is well recognized that calcium is essential for preventing osteoporosis. The positive role of micronutrients in bone metabolism is dual: minerals can impact hydroxyapatite formation or can function as regulators and cofactors [86]. Three proteins belonging to the mineral absorption pathway were more abundant in osteodifferentiated cells (Table 1) (ferritin heavy chain, STEAP2 metalloreductase and metallothionein 1E, 1.77, 2.60, and 6.56-fold increase, respectively). Metallothionein 1E (MT1E), a metal-binding protein, has different roles. For example, it maintains the concentrations of free intracellular zinc/copper ions and acts as an antioxidant. Specifically, one of the probable functions is controlling bone differentiation through regulating the availability of zinc inside the cells [87]. Lin et al. reported that ZnCl_2_ promoted DPSCs differentiation through upregulation of MT1E [88]. We revealed the massive abundance of MT1E in osteoblasts confirming its essential role in osteogenesis. Iron overload can inhibit osteodifferentiation due to ferritin overexpression [89]. In contrast, yet in line with our results, Hu et al. reported more abundant ferritin heavy chain in osteodifferentiated cell [90]. Our data suggest a high potential of nutritional therapeutics for bone disorders.

Summing up, FoxO signaling may prevent apoptosis by inducing antioxidants, thus counterbalancing proinflammatory chemokines from the IL-17 pathway. Hence, differentially abundant chemokines and matrix metallopeptidases from IL-17 pathway are potential pharmacology targets for preventing osteoclastogenesis. Thyroid-stimulating hormone might promote differentiation of osteoblasts through activation of its receptors and accumulated transporter. Finally, the accumulation of proteins belonging to the mineral absorption pathway suggests a high potential for nutritional therapy of bone disorders.

#### 2.4.2. Lipid Metabolism and PPAR Signaling Are Plausibly Connected to Energy Metabolism in the Context of Bone Development

*PPAR signaling pathway*. The nuclear peroxisome proliferator-activated receptors (PPARs) regulate many physiological processes, including adipogenesis, lipid metabolism, insulin sensitivity, inflammation, angiogenesis, and osteoclastogenesis. Additionally, they are key modulators of skeletal remodeling. PPARs belong to the superfamily of ligand-inducible nuclear hormone receptors, which includes three major members [91] with different expression profiles: (i) PPAR*α* participates in fatty acid metabolism and lowers level of lipids, (ii) PPAR*β*/*δ* mainly influence fatty acid oxidation and transport, while (iii) PPAR*γ* is primarily involved in the regulation of the adipogenesis and energy balance [91]. We identified differentially accumulated proteins from cascades of all three PPAR members (Table 1). PPAR*γ* plays a negative role in osteoblast differentiation by operating in an adipogenesis-dependent manner due to the relationship between osteoblast and adipocyte differentiation. When PPAR*γ* expression was ablated in mature osteoblasts and osteocytes, the mutant mice exhibited increased femoral bone mineral density and trabecular bone volume as well as reduced fat mass and higher energy expenditure [92]. In contrast, pharmacological induction of PPAR*α* expression concomitantly enhanced expression of bone morphogenetic protein 2 (BMP-2) and calcium deposition [93]. We identified accumulation of ApoA2 and ApoC3 (5.14 and 2.18-fold increase, respectively) working downstream of PPAR*α* pathway in osteoblasts, the former additionally validated by immunocytochemistry (Figure 3). Of note, it was reported that ApoA1 deficient mice developed osteoarthritis [94]. Information about ApoA2 and ApoC3 impact on osteogenesis is rare. ApoA2 constitutes roughly 20% of HDL cholesterol and stabilizes its structure, whereas ApoA1 is a major cholesterol-binding protein. Histological studies indicated that both mature osteoblasts and differentiating osteoblast progenitors contain stored lipids. Moreover, in vivo and in vitro studies showed that osteoblasts uptake circulating lipoproteins and free fatty acids [95].

*Fatty acid biosynthesis and degradation*. Besides peroxisomes, fatty acid biosynthesis and degradation take place in mitochondria. These pathways include acetyl-CoA acyltransferase 2, enoyl-CoA delta isomerase 2, enoyl-CoA hydratase short chain 1, acyl-CoA synthetase long chain family member 3, acyl-CoA synthetase long chain family member 4, hydroxyacyl-CoA dehydrogenase, acyl-CoA oxidase 1, acyl-CoA synthetase 2, and acyl-CoA oxidase 3 (1.06, 3.70, 1.45, 2.17, 2.03, 1.16, 2.41, 2.13, 3.11-fold increase, respectively). Overall, fatty acids represent one of the major energy sources involved in bone turnover. Furthermore, van Gastel et al. [96] proposed a role for fatty acid utilization during fracture healing and the determination of skeletal cell fate. In addition, the oxidation of fatty acids has an impact on bone metabolism and calcification [97]. Energy role of fatty acids on the mineralization phase of osteodifferentiation is further supported by the accumulation of amino acids valine, isoleucine, and threonine (discussed in Section 2.5), which could be precursors for lipogenesis through TCA cycle. Of note, our results are in line with the proteomic study of Wang et al. [98]. These authors reported a more prominent PPAR signaling and fatty acid biosynthesis pathway in in vitro osteoinduced DPSCs comparing to periodontal ligament stem cells.

*Thermogenesis*. Although thermogenesis pathway was not enriched, its identification underlined the context of molecular mechanism in osteoblast. The term thermogenesis primarily refers to the production of heat for maintaining body temperature (thermoregulation). It is also widely used to describe the regulation of the energy balance in the whole body [99]. We detected eleven members of thermogenesis pathway, accumulated in differentiated cells (Table 1). Particularly, proteins participating in energy production, ACSL3 and ACLS4 (2.17, 2.03-fold increase, respectively) as members of β-oxidation, ATP5MG (2.11-fold increase) responsible for ATP synthesis in mitochondria, NDUFS1, NDUFS3, NDUFV2, and NDUFAF2 (1.02, 1.01, 1.1, 2.97-fold increase, respectively) involved in electron transport chain in mitochondria, SDHA and SDHB (1.13, 1.47-fold increase, respectively) enzymes of TCA cycle. Bone remodeling is an energy-consuming process due to the synthesis of extracellular matrix proteins and the accumulation of mineral ions for hydroxyapatite crystal formation [95]. Interestingly, the authors reported that insulin signaling affected both bone metabolism and energy production [59]. Energy is harvested from different sources (glucose, glutamine, fatty acids, as well as autophagy) but the role of a particular substrate on the specific phase of the osteodifferentiation is not completely clear yet. Osteoclasts’ differentiation and their fusion are likely accompanied by oxidative phosphorylation [100], while another study reported increased oxidative phosphorylation during the differentiation of osteoblasts from bone marrow stem cells [101]. Our data pointed to fatty acids as major sources for energy production in the mineralization phase of osteodifferentiation.

Our data suggested the essential role of lipids on the mineralization phase of osteodifferentiation: (i) apolipoproteins A2 and C3 from PPARα cascade together with cholesterol biosynthesis enzyme DHCR7 stabilize HDL cholesterol and (ii) accumulated proteins involved in biosynthesis and degradation of fatty acids ensure preferential catabolism of lipids via thermogenesis pathway.

### 2.5. Pathways Potentially Essential for Osteodifferentiation Hinted by Metabolomic Data

Using metabolomics, we identified eight statistically significant differentially regulated metabolites, of which seven were amino acids and one acetate. All metabolites and/or metabolic pathways were upregulated in osteodifferentiated cells (Figure 7 and Appendix A). Only aminoacyl-tRNA biosynthesis and valine, leucine, isoleucine metabolism reported more than one metabolite incorporated in the pathway (Appendix A).

*Aminoacyl-tRNA synthesis*. Genetic information is transformed into functional proteins through transcription and translation. Aminoacyl-tRNA synthesis plays a key role in determining how the genetic code is interpreted [102]. This metabolic pathway determined on the 24th day of osteodifferentiation is upregulated in differentiated cells. Affected protein synthesis was also reported by Klontzas et al. in a metabolomic study of osteodifferentiation [103]. We detected aminoacyl-tRNA biosynthesis at both levels, by incorporating proteins and metabolites into functional pathways. We identified two members on protein level (histidyl-tRNA synthetase and glutaminyl-tRNA synthetase, 4.20 and 2.0-fold increase in differentiated cells, respectively) and seven members by metabolomic analysis (histidine, phenylalanine, valine, isoleucine, threonine, tryptophan, and proline; 2.3, 2.96, 3.5, 4.0, 1.4, 4.8, and 2.9-fold increase, respectively). Both substrate and enzyme were determined only in the case of histidine and histidyl-tRNA synthetase, which may indicate their prominent role in the osteogenic process.

*Histidine and phenylalanine metabolism*. Histidyl-tRNA synthetase is necessary for the synthesis of histidyl-transfer RNA, which is essential for histidine incorporation into proteins [104]. HIStidin-rich proteins also include histatins, proteins with various function, among other with wound-healing properties. For example, histatins were used for the treatment of oral diseases [105]. It was reported that human salivary Hst1 protein promoted BMP-2 induced bone regeneration [106]. In our study, BMP-2 protein was detected (complementation) both by Western blotting and immunocytochemistry (Figure 3 and Figure 6). We also determined histidine as a solitary member of histidine and beta-alanine metabolism (2.3-fold increase in osteodifferentiated cells). Histidine metabolism contains several pathways. The most important is catabolism via urocanate to glutamate, relevant for HIS-rich proteins and HIS-containing dipeptides synthesis [105], especially L-carnosine (L-Car). L-Car is synthetized from L-histamine and beta-alanine and its zinc chelated form β-alanyl-L-histidinato zinc is a pharmacological tool for treatment of osteoporosis [107]. Chauhan and Singh reported that the presence of histidine led to hydroxyapatite mineralization with plate-like morphology in studies with mouse osteoblast precursor cells [108]. Hydroxyapatite is a basic building block of bones and teeth. These findings correspond to our observations. On the 24th day after application of osteogenic medium differentiated cells reached mineralization phase accompanied with production of calcium deposits (Figure 4). Aromatic amino acid phenylalanine (2.96-fold increase in differentiated cells) is a member of three metabolic pathways. The authors showed that supplementation of C57BL/6 mice model by aromatic amino acids (phenylalanine, tyrosine, and tryptophan) prevented bone loss [109].

*Valine, leucine, and isoleucine biosynthesis and degradation*. As we described in part thermogenesis, bone remodeling is energy consuming process and the involvement of certain amino acids in metabolism can bring another source of energy. Isoleucine, valine, and threonine (4.0, 3.5, and 1.4-fold increase, respectively) are three from nine essential amino acids. Their products can enter TCA cycle for energy production or they can be precursors for lipogenesis and/or ketone bodies production through acetyl-CoA and acetoacetate [110].

In summary, analysis of metabolites indicated their plausible support for protein synthesis and energy metabolism.

### 2.6. Proposed Proteins and Metabolites for Functional Validation in Patient Samples

Although Wang et al. [98] argued that different stem cell sources might employ different major pathways for the osteodifferentiation, we addressed this limitation by integrating and discussing our data in the context of published literature on different biological material and experimental approaches. Hence, we propose that the particular proteins or metabolites in pathways with high enrichment score are potential targets for validation in patient samples. Moreover, we suggest that several proteins/metabolites involved in osteogenesis and/or bone loss belonging to discussed pathways could be promising biomarkers. These are: (i) IP3R, histidine, decorin, and inhibin β_A_ from known pathways; and (ii) apolipoproteins (ApoA2/C3), 7-dehydrocholesterol reductase, matrix metallopeptidases (MMP1/3), chemokines (CXCL1/5/6), metallothionein 1E, signal transducers and activators of transcription (STAT1/3), and PRMT1 from enriched pathways.

## 3. Materials and Methods

### 3.1. Cell Cultures and Osteogenic Treatment

The human commercial cell line, DPSCs (Lonza, Basel, Switzerland) was used. For all studies examining the molecular mechanism of osteogenic differentiation, cells were seeded in 25 cm^2^ flasks and grown under a humidified atmosphere of 5% CO_2_ at 37 °C. Cells were precultured in DPSCs growth medium consisted of α-MEM (Merck Sigma-Aldrich, Darmstadt, Germany) supplemented with 2 mM GlutaMAX (Thermo Fisher Scientific, Waltham, MA, USA), 0.2 mM ascorbic acid 2-phosphate (Sigma-Aldrich), 10% fetal bovine serum (Biosera, Nuaille, France), 100 U/mL penicillin, and 100 µg/mL streptomycin (Biosera). The medium was replaced every three days. During preculture, cells remained in an undifferentiated stage. After reaching 80–90% confluence, cells were harvested using TrypLE Express Enzyme (Thermo Fisher Scientific) and then seeded at 5000 cells/cm^2^. When DPSCs reached 50% confluence (day 0), the growth medium was replaced with osteoinductive medium OsteoMAX-XF (Merck Sigma-Aldrich) for osteogenic differentiation and mineralization. The cells were grown in this specific medium for 24 days. In parallel, undifferentiated DPSCs (control) were cultured in the DPSCs growth medium. Media were changed every 3 days. Cell morphology was monitored during the experiment by light microscopy (Optika, Via Rigla, Italy).

### 3.2. Alizarin Red S Staining and Quantification of Calcium Mineral Deposits

Control and differentiated cells were grown in 24 well plates. On a specific day (5th, 11th, 15th, 20th, 24th) after osteogenic medium application, cells were fixed by 4% paraformaldehyde (Cell Signaling Technology, Danvers, MA, USA) for 30 min at RT. After DPBS (Gibco, Paisley, UK) washing, formed Ca^2+^ deposits were stained by 2% Alizarin Red S (Merck Sigma-Aldrich, Darmstadt, Germany) for 3 min at RT. Next, the dye was aspired and cells were carefully washed three times by ultrapure water to remove the remaining stain. Finally, cells were kept in ultrapure water. Color intensity was monitored and images were captured by the inverted microscope XDS-2 (Optika). For quantification of total stained mineralized tissue, Alizarin Red S was eluted by destain solution (10% acetic acid, 20% methanol) for 15 min at RT [111]. The eluates were centrifuged (10,000× *g*) for 10 min at RT to remove mineral precipitates and cell debris. Then, the supernatant was transferred into 96 well plates in duplicates. The absorbance was measured at 405 nm using Synergy H1 Microplate Reader (BioTek, Winooski, VT, USA). The amount of mineral deposits is presented as optical density normalized to the total amount of proteins (mg), which were determined by BCA assay (Thermo Fisher Scientific).

### 3.3. Immunocytochemistry

Immunocytochemistry was performed according to the modified protocol [21]. Briefly, cells were seeded in lumox 96-multiwell plates (Sarstedt, Nümbrecht, Germany). On the 24th day after osteogenic induction, cells were washed with DPBS (Gibco) and fixed with 4% paraformaldehyde (Cell Signaling Technology) for 30 min at room temperature (RT). Then, fixed cells were washed with DPBS for 10 min, permeabilized with 0.2% Triton-X in DPBS (washing buffer, WB) and blocked with 5% goat serum in WB for 1 h at RT. Consequently, samples were incubated with appropriately diluted primary antibodies in WB at 4 °C overnight. Then, the cells were washed with WB three times for 10 min and further incubated with goat anti-rabbit Alexa Fluor 488-conjugated IgG (1:500, #ab150077, Abcam, Cambridge, UK) or goat anti-mouse Alexa Fluor 488-conjugated IgG (1:500, #ab150113, Abcam) secondary antibody in WB for 1 h at RT in the dark. After washing, nuclei were stained with DAPI (1 µg/mL) for 10 min at RT in the dark. Cells were again washed as described above, cellular proteins and nuclei were visualized by fluorescent microscope WiScan (Hermes IDEA Bio-Medical, Rehovot, Israel), magnification 100×.

The following primary antibodies were used: anti-bone morphogenetic protein 2, BMP-2 (1:100, #ab14933, Abcam), anti-apolipoprotein A2, ApoA2 (1:50, #ab92478, Abcam), anti-bone sialoprotein, BSP II (1:50, #5468, Cell Signaling Technology), and anti-osteocalcin, OSTCN (1:80, #ab13418, Abcam).

### 3.4. Protein Extraction

Control and differentiated cells were harvested in biological triplicates on the 24th day after application of the osteogenic medium. Cells were washed with ice-cold DPBS (Gibco) and scraped-off from flasks using SDS buffer (4% SDS, 0.1 M Tris pH 7.6, 100 mM DTT) supplemented with protease (Roche, Basel, Switzerland) and phosphatase inhibitors (Sigma-Aldrich). Then, lysates were transferred to the tubes, incubated at 10 °C for 15 min and sonicated (3 × 10 s, 20% pulse) followed by 20 min centrifugation (14,000× *g*) at 12 °C. The final supernatant was aliquoted and stored at −80 °C until use. Protein concentration was determined by the Bradford detergent compatible assay (Thermo Fisher Scientific). Control and differentiated cells were processed in parallel.

### 3.5. SDS-PAGE and Western Blotting

Equal amounts (40 µg) of protein aliquots in the Laemmli buffer (BioRad, Hercules, CA, USA) from both control (undifferentiated cells) and osteoblasts (differentiated cells) were loaded on Tris-glycine gradient (4–20%) or 12% polyacrylamide gels (BioRad) and electrophoretically separated in Mini-PROTEAN Tetra cell chamber (BioRad). Separation was performed at 60 V for 20 min, followed by 200 V until tracking dye reached the bottom of the gel. Then, proteins profiled in the gel were transferred to a nitrocellulose membrane (Thermo Fisher Scientific) using a semi-dry blotting unit (Biometra, Jena, Germany) and transferring condition 1 mA/cm^2^. The membrane was blocked (5% non-fat dry milk or 5% BSA in Tris-buffered saline with Tween 20, TBS-T) for 1 h at room temperature and incubated overnight with primary antibodies to target or reference proteins at 4 °C. Antibodies were diluted in 5% non-fat dry milk or 5% BSA. After incubation, the membrane was thoroughly washed with TBS-T three times and further incubated with horseradish peroxidase-conjugated goat anti-rabbit IgG (1:2000, #7074, Cell Signaling Technology) or horseradish peroxidase-conjugated goat anti-mouse IgG (1:2500, #ab6789, Abcam) for 2 h at RT. Proteins were detected by chemiluminescence after application of the developing solution according to manufacturer’s instructions (#34080, Thermo Fisher Scientific) and visualized using ChemiDoc XRS+ system (BioRad). Densitometric analysis of protein abundance was accomplished via ImageLab software (version 6.0) (BioRad). Signals were normalized to glyceraldehyde 3-phosphate dehydrogenase (GAPDH).

The following primary antibodies were used: anti-dentin sialophosphoprotein, DSPP (1:500, #PA5-76382, Invitrogen, Waltham, MA, USA), anti-dentin matrix protein 1, DMP-1 (1:500, #PA5-103323, Invitrogen), anti-bone morphogenetic protein 2, BMP-2 (1:250, #ab14933, Abcam), anti-collagen type VI alpha-1 chain, Col6A1 (1:250, #sc-377143, Santa Cruz Biotechnology, Dallas, TX, USA), anti-inositol 1,4,5-trisphosphate receptor, IP3R (1:250, #8568, Cell Signaling Technology), and anti-GAPDH (1:500, #sc-166574, Santa Cruz Biotechnology).

### 3.6. Protein Digestion by Filter-Aided Sample Preparation

Total protein extracts (50 µg) adjusted to 200 µL with urea buffer (8 M urea in 50 mM ammonium bicarbonate) were digested using filter-aided sample preparation [112] on Microcon Ultracel YM-10 centrifugal filters (Merck Sigma Aldrich, Darmstadt, Germany). Before digestion, the filters were pre-washed by 50 mM ammonium bicarbonate. Samples were incubated overnight at 37 °C with trypsin gold (Promega, Madison, WI, USA) in the proportion 1:75 to total proteins and consequently slightly acidified by 10% trifluoroacetic acid. The peptide mixtures were purified on Sep-Pak C18 light columns (Waters, Millford, MA, USA) and vacuum-concentrated. The concentration of peptides was measured spectrophotometrically by NanoDrop 2000 (Thermo Fisher Scientific) and adjusted to 100 ng/μL with 0.1% trifluoroacetic acid.

### 3.7. Relative Label-Free Quantification by Mass Spectrometry

Complex peptide mixtures (400 ng) were separated using Acquity M-Class UHPLC (Waters). Firstly, samples were desalted and concentrated on the nanoEase Symmetry C18 trap column: 20 mm length, 180 μm diameter, 5 μm particle size (Waters). Secondly, peptides were comprehensively profiled on the nanoEase HSS T3 C18 analytical column: 250 mm length, 75 μm diameter, 1.8 μm particle size (Waters) using a 118 min gradient of 5–35% acetonitrile with 0.1% formic acid at a flow rate of 300 nL/min. Next, samples were nanosprayed by 2.9 kV applied to PicoTip emitter (New Objective, Woburn, MA, USA) to the quadrupole time-of-flight mass spectrometer Synapt G2-Si (Waters). The instrument operated in resolution mode with enabled ion mobility, which provided additional gas-phase separation of ions, particularly useful for filtering non-peptide contaminants. Synapt G2-Si was tuned using parameters described in the literature [112] for maximum sensitivity and both mass and mobility resolutions. Spectra were recorded in a data-independent mode referred as high definition MSE. Ions with 50–2000 *m*/*z* were detected in both low energy and high energy channels. Voltage profile setting on the first quadrupole allowed efficient deflection of low mass precursor ions ˂ 400 *m*/*z*. The external standard Glu1-Fibrinopeptide B was infused for mass correction.

Compression and Archival Tool 1.0 (Waters) reduced noise, removing ion counts below 15. Subsequent processing was done in Progenesis QI 4.1 (Waters). Peaks were modeled with low energy threshold 400 counts and high energy threshold 40 counts. Chromatographic elution profiles in low and high energy traces were correlated to assign fragment ions to appropriate precursor ions. Retention times of peaks were aligned to automatically selected reference chromatogram. Next, peak intensities were normalized to the median distribution of all ions. The label-free quantification relied on the integration of peak areas of the three most intense precursor peptides. Arcsinh transformation of data was done before statistics. We used Ion Accounting 4.0 (Waters) search algorithm for protein identification. Spectra were searched against human proteome sequences downloaded from UniProt in September 2019 (74,823 entries, uniprot.org). Search parameters were: (i) one trypsin miscleavage, (ii) fixed carbamidomethyl cysteine, (iii) variable oxidized methionine and deamidated asparagine/glutamine, (iv) automatic mass tolerance, (v) 4% false discovery rate against the randomized database. Protein hits having unique peptides were accepted if at least two reliable peptides (score ≥ 5.93, mass accuracy ≤ 20 ppm) matched the sequence. Integrated principal component analysis visualized clustering of biological replicates according to experimental groups.

### 3.8. Metabolite Extraction

Control and differentiated cells were harvested in biological triplicates on the 24th day after application of osteogenic medium according to the modified protocol [113]. Briefly, cells were washed with cold DPBS (Gibco) and scraped-off from the flask using 90% methanol. Then, lysed cells were transferred to the tubes, incubated 15 min at −20 °C followed by 20 min centrifugation (16,000× *g*) at 4 °C. The final supernatant was stored at −80 °C until use. Control and differentiated cells were processed in parallel.

### 3.9. Identification and Quantification of Metabolites by Nuclear Magnetic Resonance

The supernatant was vacuum-dried and then the residue was dissolved in 550 µL of 200 mM phosphate buffer pH 7.4, containing 0.2 mM TMS-d_4_ (trimethylsilylpropionic acid-d_4_) as chemical shift reference, in deuterated water.

Data were acquired on 600 MHz NMR spectrometer Avance III (Bruker, Bremen, Germany) equipped with cryoprobe at 310 K. Samples were freshly prepared before acquisition and measured in random order. Before measurement, each sample was prewarmed to 310 K for 5 min. An exponential noise filter was used to introduce 0.3 Hz line broadening before Fourier transformation. The proton NMR chemical shifts are reported relative to TMS-d_4_ signal and data were zero filled once. For each sample, one-dimensional as well as two-dimensional NMR spectra were acquired. We modified standard profiling protocol (Bruker) as follows: (i) noesy with presaturation (FID size 64 k, dummy scans 4, number of scans 4, spectral width 20.4750 ppm); (ii) cosy with presaturation (FID size 4 k, dummy scans 8, number of scans 8, spectral width 16.0125 ppm); (iii) homonuclear j-resolved (FID size 8 k, dummy scans 16, number of scans 64, spectral width 16.6087); (iv) profiling cpmg (FID size 64 k, dummy scans 4, number of scans 2048, spectral width 20.0156 ppm). All experiments were conducted with a relaxation delay of 4 s.

All spectra were binned from 0.00 ppm to 10.00 ppm with bins 0.001 ppm. The multiplicity of peaks was confirmed in j-resolved spectra, and homonuclear cross peaks were confirmed in cosy spectra. Spectra were solved using Human Metabolome Database (www.hmda.ca, accessed on 15 September 2019) [114], internal metabolite database, Chenomx software (version 8.31), and metabolomic literature [115]. After identification of metabolites, we chose spectra subregions with unique assignments or minimally affected by interfering signals. In 0.001 ppm binned spectra, we summed integrals of selected metabolites. These data represent a relative concentration of a particular metabolite in the sample. Metabolites showing peaks with low intensity or strong peaks´ overlap were excluded from the evaluation. Next, metabolites´ relative abundances were normalized to the total amount of proteins [116].

### 3.10. Statistical Analysis and Bioinformatics

Differentially abundant proteins were filtered on effect size (at least 1-fold of log2 transformed ratio) and ANOVA *p* ≤ 0.01 (false discovery rate *Q* ˂ 1.05%). For statistical evaluation of metabolomic and densitometry data, we used Student’s *t*-test at adjusted (Benjamini-Hochberg correction) *p* ≤ 0.05 with the same effect size threshold. Differentially abundant proteins were sorted into biological processes using GOTermFinder (version 0.86) [117] and metabolic pathways by KEGG (release 97.0) [118]. Proteins with identification numbers absent in KEGG were queried for homology using BLASTP. Differentially abundant metabolites were uploaded in MetaboAnalyst (version 5.0) for functional analysis [119]. Pathway enrichment analysis was performed in R programming language (version 4.0.3) by ClusterProfiler package (version 3.18.1) [120] of R/Bioconductor (version 3.11) using a permutation test (*p* ≤ 0.05) without stringent false discovery rate filtering as discussed in specialized literature [121]. Dot plot was generated by enrichplot (version 1.10.2) R/Bioconductor package. A volcano plot was generated using VolcaNoseR (version 1.0.3) [122].

## 4. Conclusions

Understanding of molecular mechanisms which participate in various diseases and normal development is necessary both for expanding fundamental knowledge and for developing new treatment strategies. Complex proteomic and metabolomic data allowed us to identify important proteins and metabolites and to suggest their role in the osteodifferentiation, summarized in Figure 8. In essence, we explained the molecular mechanisms behind late osteoblasts’ differentiation (mineralization phase) by discussing several known (described in the literature) as well as unconventional signaling pathways in relation to osteogenesis. Our proteomic data indicated that the osteodifferentiation in the mineralization phase is energy consuming, whereas the cells harvested energy probably from fatty acids. Furthermore, this process likely required intensive antioxidant protection and was plausibly regulated by hormones. Metabolomic data enriched results by highlighting pathways related to energy metabolism and protein synthesis. Of note, the presented network of signaling pathways is largely hypothetical and should be validated in follow-up functional experiments. For instance, inhibitors of particular signaling pathways may be used for direct verification if specific molecular mechanisms indeed promote or inhibit osteodifferentiation of mesenchymal stem cells.

## Figures and Tables

**Figure 1 ijms-22-07908-f001:**
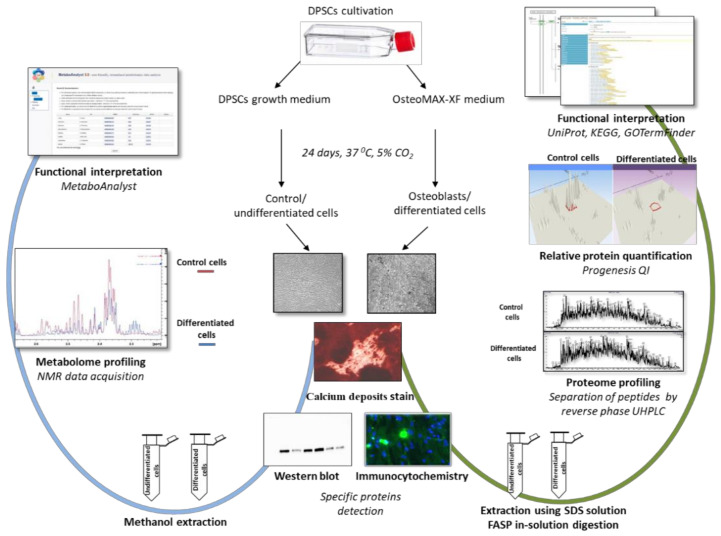
Experimental design—analytical workflow. We followed undifferentiated and differentiated human dental pulp stem cells (DPSCs). Osteogenic differentiation was verified by calcium deposits stain, Western blotting, and immunocytochemistry. Proteins extracted from three biological replicates were digested using filter-aided sample preparation (FASP). Following that, we directly quantified proteins by mass spectrometry upon comprehensive two-dimensional separation of analytes. Metabolites were directly quantified by nuclear magnetic resonance (NMR). Dedicated KEGG, GOTermFinder, and MetaboAnalyst tools accompanied with enrichment analysis were used for revealing the biological meaning of the extensive dataset.

**Figure 2 ijms-22-07908-f002:**
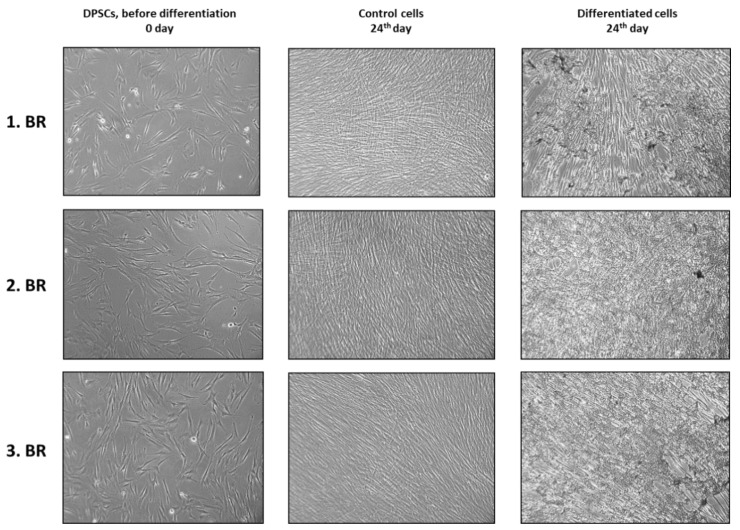
The visual appearance of undifferentiated (control) and differentiated cells on the 24th day after osteogenic medium application. Controls were cultured in a DPSCs growth medium during the experiment. Cellular morphology correlates with the osteogenic differentiation potential. Representative micrographs were taken by phase contrast technique at magnification 100×.

**Figure 3 ijms-22-07908-f003:**
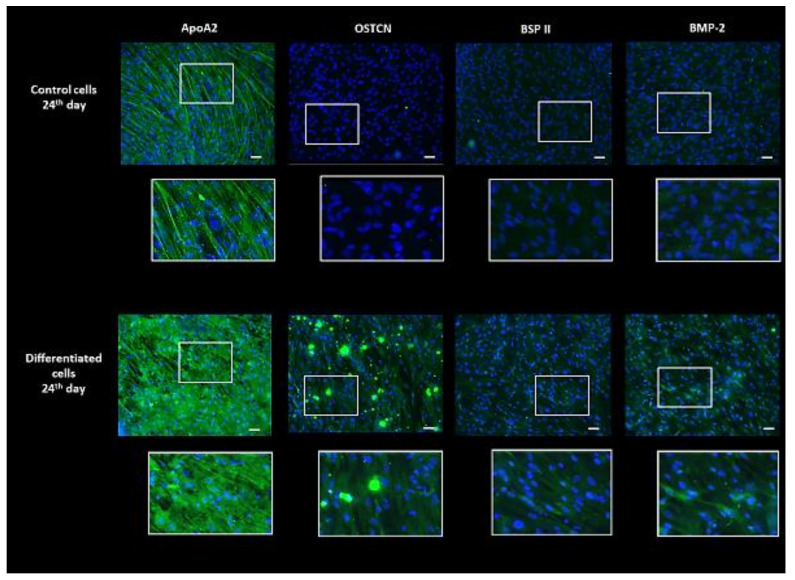
Content of osteogenic markers and apolipoprotein A2. Cells were fixed with 4% paraformaldehyde. Specific proteins apolipoprotein A2 (ApoA2), osteocalcin (OSTCN), bone sialoprotein (BSP II), and bone morphogenetic protein (BMP-2) were detected by immunocytochemistry using WiScan microscope. Nuclei were visualized by DAPI stain. Bar = 50 µm. Areas with white rectangles are shown as magnified insets below each micrograph.

**Figure 4 ijms-22-07908-f004:**
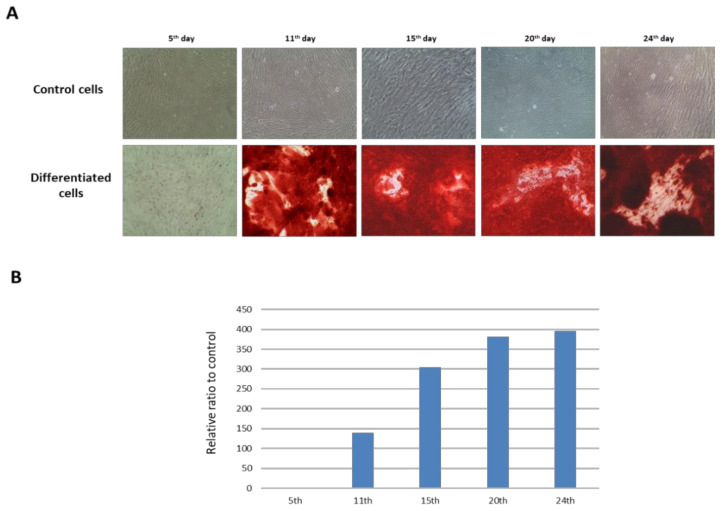
Morphological changes of DPSCs during osteogenic differentiation on the 5th, 11th, 15th, 20th, and 24th day and quantitative evaluation of Ca^2+^ deposits. The control cells (upper line) and osteodifferentiated cells (lower line) were stained by 2% Alizarin Red S and monitored by light microscopy at magnification 100× (**A**). The mineral deposits production was quantified at each time point spectrophotometrically and represented as a relative ratio to control cells (**B**).

**Figure 5 ijms-22-07908-f005:**
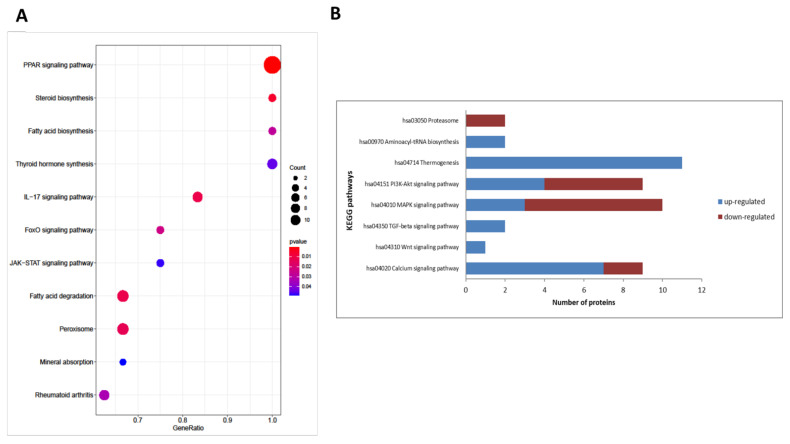
KEGG pathway analysis. Dot plot (**A**) of enriched KEGG pathways (results from ClusterProfiler) produced by enrichplot Bioconductor package. The color of the dots represents the *p*-value, the size of the dots represents the number of differentially abundant proteins. Column chart (**B**) of KEGG pathway with a known role in osteogenesis. Specific proteins belonging to these functional groups are detailed in Table 1.

**Figure 6 ijms-22-07908-f006:**
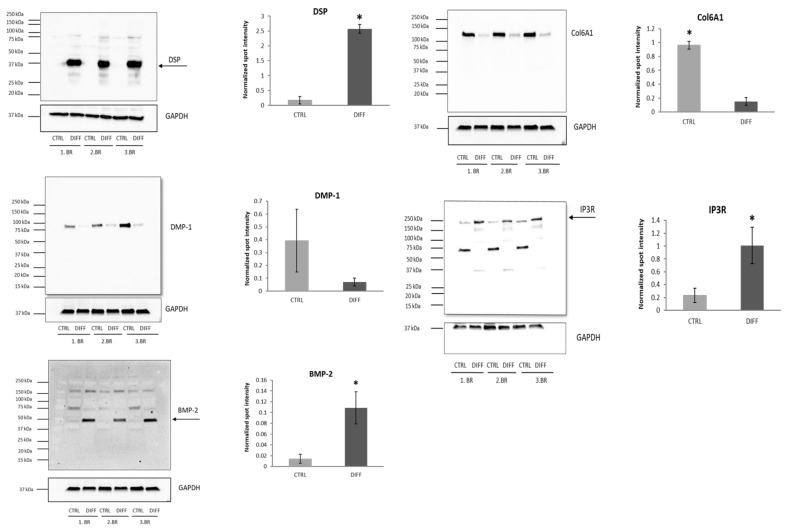
Confirmation of several differentially abundant proteins by Western blotting. Densitometric analysis of protein abundance was performed using ImageLab software. Values represent the means of three biological replicates. Error bars denote the standard deviations, * indicate *p* ≤ 0.05. Arrows indicate densitometrically analyzed major protein bands: DSP a likely cleavage product of DSPP, BMP-2 mature protein, and functional IP3R tetramer. Abbreviations: DSP (dentin sialoprotein), DMP-1 (dentin matrix protein 1), BMP-2 (bone morphogenetic protein 2), IP3R (inositol 1,4,5-trisphosphate receptor), Col6A1 (collagen 6 alpha 1 chain), GAPDH (glyceraldehyde 3-phosphate dehydrogenase).

**Figure 7 ijms-22-07908-f007:**
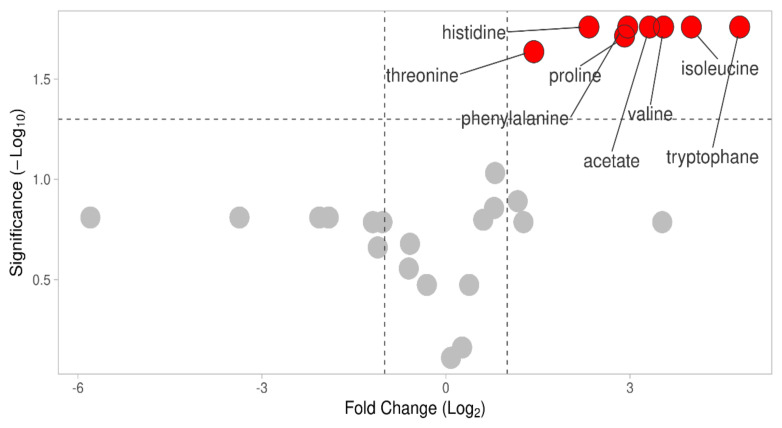
Volcano plot of metabolomic data. Differentially abundant metabolites are highlighted.

**Figure 8 ijms-22-07908-f008:**
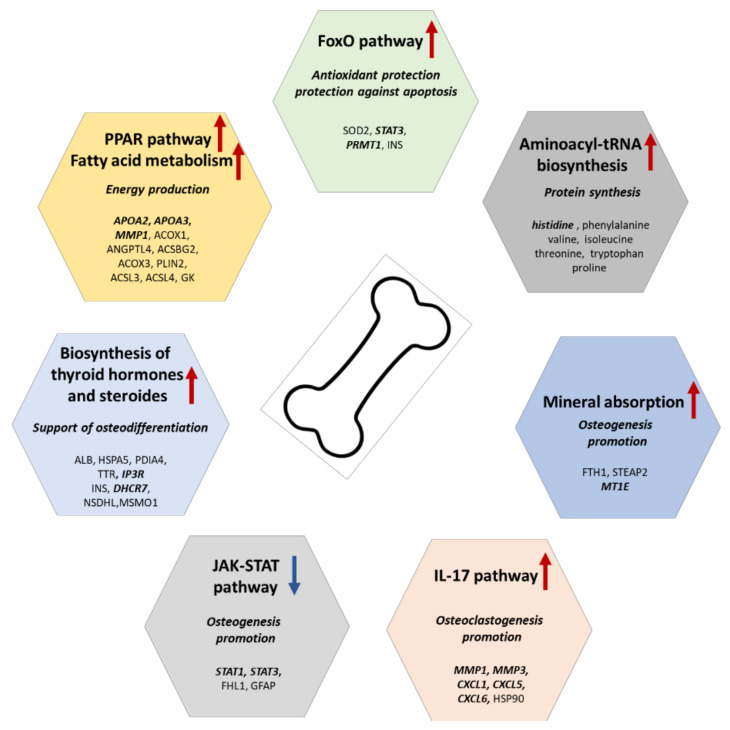
Summary of the proposed molecular mechanisms of osteodifferentiation. Potential biomarkers are bold and italic denoted. Abbreviations: ALB (albumin), ACSBG2 (acyl-CoA synthetase bubblegum family member 2), ACSL1, ACSL3 (acyl-CoA synthetase long chain family members 1 and 3), ANGPTL4 (angiopoietin like 4), ACOX1, ACOX3 (acyl-CoA oxidases 1 and 3), APOA2, APOC3 (apolipoproteins A2 and C3), CXCL1, CXCL5, CXCL6 (C-X-C motif chemokine ligands 1, 5, and 6), FHL1 (four and a half LIM domains 1), FTH1 (ferritin heavy chain 1), GFAP (glial fibrillary acidic protein), GK (glycerol kinase), HSP90 (heat shock protein 90), HSPA5 (heat shock protein family A (Hsp70) member 5), INS (insulin), IP3R (inositol 1,4,5-trisphosphate receptor type 2), MMP1, MMP3 (matrix metallopeptidases 1 and 3), MSMO1 (methylsterol monooxygenase 1), MT1E (metallothionein 1E), NSDHL (NAD(P) dependent steroid dehydrogenase-like), PDIA4 (protein disulfide isomerase family A member 4), PLIN2 (perilipin 2), PRMT1 (protein arginine methyltransferase 1), SOD2 (superoxide dismutase 2), STAT1, STAT3 (signal transducers and activators of transcription 1 and 3), STEAP2 (STEAP2 metalloreductase), and TTR (transthyretin).

**Table 1 ijms-22-07908-t001:** List of proteins belonging to KEGG pathways discussed in the text and visualized in Figure 5.

KEGG Identifier	Symbol	Protein Name	UniProt Identifier	Log2 Ratio Diff./Control
**Ranked list of relevant enriched pathways**				
**hsa03320 PPAR signaling pathway** (**11**), ***p*-value 0.00075**				
hsa:123	PLIN2	perilipin 2	Q99541	1.50
hsa:2181	ACSL3	acyl-CoA synthetase long chain family member 3	O95573	2.17
hsa:2182	ACSL4	acyl-CoA synthetase long chain family member 4	O60488	2.03
hsa:2710	GK	glycerol kinase	P32189	1.53
hsa:336	APOA2	apolipoprotein A2	P02652	5.14
hsa:345	APOC3	apolipoprotein C3	P02656	2.18
hsa:4312	MMP1	matrix metallopeptidase 1	P03956	3.09
hsa:51	ACOX1	acyl-CoA oxidase 1	Q15067	2.41
hsa:51129	ANGPTL4	angiopoietin like 4	Q9BY76	1.90
hsa:81616	ACSBG2	acyl-CoA synthetase bubblegum family member 2	Q5FVE4	2.13
hsa:8310	ACOX3	acyl-CoA oxidase 3, pristanoyl	O15254	3.11
**hsa00100 Steroid biosynthesis** (**3**), ***p*-value 0.00593**				
hsa:1717	DHCR7	7-dehydrocholesterol reductase	Q9UBM7	2.77
hsa:50814	NSDHL	NAD(P)-dependent steroid dehydrogenase-like	Q15738	2.47
hsa:6307	MSMO1	methylsterol monooxygenase 1	Q15800	2.38
**hsa04657 IL-17 signaling pathway** (**6**), ***p*-value 0.01139**				
hsa:2919	CXCL1	C-X-C motif chemokine ligand 1	P09341	1.16
hsa:3320	HSP90AA1	heat shock protein 90 alpha family class A member 1	P07900	1.99
hsa:4312	MMP1	matrix metallopeptidase 1	P03956	3.09
hsa:4314	MMP3	matrix metallopeptidase 3	P08254	4.04
hsa:6372	CXCL6	C-X-C motif chemokine ligand 6	P80162	2.67
hsa:6374	CXCL5	C-X-C motif chemokine ligand 5	P42830	2.04
**hsa00071 Fatty acid degradation** (**9**), ***p*-value 0.01192**				
hsa:10449	ACAA2	acetyl-CoA acyltransferase 2	P42765	1.06
hsa:10455	ECI2	enoyl-CoA delta isomerase 2	O75521	3.70
hsa:1892	ECHS1	enoyl-CoA hydratase, short chain 1	P30084	1.23
hsa:2181	ACSL3	acyl-CoA synthetase long chain family member 3	O95573	2.17
hsa:2182	ACSL4	acyl-CoA synthetase long chain family member 4	O60488	2.03
hsa:3030	HADHA	hydroxyacyl-CoA dehydrogenase trifunctional multienzyme complex sb. α	P40939	1.16
hsa:51	ACOX1	acyl-CoA oxidase 1	Q15067	2.41
hsa:81616	ACSBG2	acyl-CoA synthetase bubblegum family member 2	Q5FVE4	2.13
hsa:8310	ACOX3	acyl-CoA oxidase 3, pristanoyl	O15254	3.11
**hsa04146 Peroxisome** (**9**), ***p*-value 0.01345**				
hsa:10455	ECI2	enoyl-CoA delta isomerase 2	O75521	3.70
hsa:10901	DHRS4	dehydrogenase/reductase 4	Q9BTZ2	1.14
hsa:1891	ECH1	enoyl-CoA hydratase 1	Q13011	1.45
hsa:2181	ACSL3	acyl-CoA synthetase long chain family member 3	O95573	2.17
hsa:2182	ACSL4	acyl-CoA synthetase long chain family member 4	O60488	2.03
hsa:3418	IDH2	isocitrate dehydrogenase (NADP(+)) 2	P48735	1.21
hsa:51	ACOX1	acyl-CoA oxidase 1	Q15067	2.41
hsa:6648	SOD2	superoxide dismutase 2	P04179	3.17
hsa:8310	ACOX3	acyl-CoA oxidase 3, pristanoyl	O15254	3.11
**hsa04068 FoxO signaling pathway** (**4**), ***p*-value 0.02370**				
hsa:3276	PRMT1	protein arginine methyltransferase 1	Q99873	2.31
hsa:3630	INS	insulin	P01308	2.64
hsa:6648	SOD2	superoxide dismutase 2	P04179	3.17
hsa:6774	STAT3	signal transducer and activator of transcription 3	P40763	−1.11
**hsa00061 Fatty acid biosynthesis (3), *p*-value 0.02800**				
hsa:2181	ACSL3	acyl-CoA synthetase long chain family member 3	O95573	2.17
hsa:2182	ACSL4	acyl-CoA synthetase long chain family member 4	O60488	2.03
hsa:81616	ACSBG2	acyl-CoA synthetase bubblegum family member 2	Q5FVE4	2.13
**hsa05323 Rheumatoid arthritis** (**8**), ***p*-value 0.03344**				
hsa:10312	TCIRG1	T cell immune regulator 1, ATPase H+ transporting V0 subunit a3	Q13488	1.25
hsa:2919	CXCL1	C-X-C motif chemokine ligand 1	P09341	1.16
hsa:4312	MMP1	matrix metallopeptidase 1	P03956	3.09
hsa:4314	MMP3	matrix metallopeptidase 3	P08254	4.04
hsa:6372	CXCL6	C-X-C motif chemokine ligand 6	P80162	2.67
hsa:6374	CXCL5	C-X-C motif chemokine ligand 5	P42830	2.04
hsa:6387	CXCL12	C-X-C motif chemokine ligand 12	P48061	2.41
hsa:9114	ATP6V0D1	ATPase H+ transporting V0 subunit d1	P61421	1.50
**hsa04918 Thyroid hormone synthesis** (**5**), ***p*-value 0.04442**				
hsa:213	ALB	albumin	P02768	2.45
hsa:3309	HSPA5	heat shock protein family A (Hsp70) member 5	P11021	1.76
hsa:3709	ITPR2	inositol 1,4,5-trisphosphate receptor type 2	Q14571	4.92
hsa:7276	TTR	transthyretin	P02766	2.51
hsa:9601	PDIA4	protein disulfide isomerase family A member 4	P13667	1.50
**hsa04630 JAK-STAT signaling pathway** (**4**), ***p*-value 0.04770**				
hsa:2273	FHL1	four and a half LIM domains 1	Q13642	−1.65
hsa:2670	GFAP	glial fibrillary acidic protein	P14136	−1.08
hsa:6772	STAT1	signal transducer and activator of transcription 1	P42224	−1.55
hsa:6774	STAT3	signal transducer and activator of transcription 3	P40763	−1.11
**hsa04978 Mineral absorption** (**3**), ***p*-value 0.04882**				
hsa:2495	FTH1	ferritin heavy chain 1	P02794	1.77
hsa:261729	STEAP2	STEAP2 metalloreductase	Q8NFT2	2.60
hsa:4493	MT1E	metallothionein 1E	P04732	6.56
**List of not enriched pathways with recognized function in osteodifferentiation**				
**hsa04020 Calcium signaling pathway** (**9**)				
hsa:2767	GNA11	G protein subunit alpha 11	P29992	−3.15
hsa:291	SLC25A4	solute carrier family 25 member 4	P12235	1.05
hsa:293	SLC25A6	solute carrier family 25 member 6	P12236	1.05
hsa:3709	ITPR2	inositol 1,4,5-trisphosphate receptor type 2	Q14571	4.92
hsa:444	ASPH	aspartate beta-hydroxylase	Q12797	1.07
hsa:7416	VDAC1	voltage dependent anion channel 1	P21796	1.18
hsa:7417	VDAC2	voltage dependent anion channel 2	P45880	1.14
hsa:7419	VDAC3	voltage dependent anion channel 3	Q9Y277	1.36
hsa:9630	GNA14	G protein subunit alpha 14	O95837	−1.99
**hsa04310 Wnt signaling pathway** (**1**)				
hsa:7474	WNT5A	Wnt family member 5A	P41221	1.24
**hsa04350 TGF-beta signaling pathway** (**2**)				
hsa:1634	DCN	decorin	P07585	1.75
hsa:3624	INHBA	inhibin subunit beta A	P08476	5.20
**hsa04010 MAPK signaling pathway** (**10**)				
hsa:1398	CRK	CRK proto-oncogene, adaptor protein	P46108	−1.08
hsa:2768	GNA12	G protein subunit alpha 12	Q03113	−1.73
hsa:3481	IGF2	insulin like growth factor 2	P01344	1.12
hsa:3630	INS	insulin	P01308	2.64
hsa:4137	MAPT	microtubule associated protein tau	P10636	−3.43
hsa:51776	MAP3K20	mitogen-activated protein kinase kinase kinase 20	Q9NYL2	−1.32
hsa:5536	PPP5C	protein phosphatase 5 catalytic subunit	P53041	−1.17
hsa:781	CACNA2D1	calcium voltage-gated channel auxiliary subunit alpha2delta 1	P54289	−1.31
hsa:929	CD14	CD14 molecule	P08571	2.42
hsa:9448	MAP4K4	mitogen-activated protein kinase kinase kinase kinase 4	O95819	−3.12
**hsa04151 PI3K-Akt signaling pathway** (**9**)				
hsa:1277	COL1A1	collagen type I alpha 1 chain	P02452	−1.19
hsa:1282	COL4A1	collagen type IV alpha 1 chain	P02462	−1.66
hsa:1291	COL6A1	collagen type VI alpha 1 chain	P12109	−1.43
hsa:3320	HSP90AA1	heat shock protein 90 alpha family class A member 1	P07900	1.99
hsa:3481	IGF2	insulin like growth factor 2	P01344	1.12
hsa:3630	INS	insulin	P01308	2.64
hsa:3912	LAMB1	laminin subunit beta 1	P07942	1.17
hsa:3913	LAMB2	laminin subunit beta 2	P55268	−1.28
hsa:5649	RELN	reelin	P78509	−1.17
**hsa04714 Thermogenesis** (**11**)				
hsa:10632	ATP5MG	ATP synthase membrane subunit g	O75964	2.11
hsa:2181	ACSL3	acyl-CoA synthetase long chain family member 3	O95573	2.17
hsa:2182	ACSL4	acyl-CoA synthetase long chain family member 4	O60488	2.03
hsa:4719	NDUFS1	NADH:ubiquinone oxidoreductase core subunit S1	P28331	1.02
hsa:4722	NDUFS3	NADH:ubiquinone oxidoreductase core subunit S3	O75489	1.01
hsa:4729	NDUFV2	NADH:ubiquinone oxidoreductase core subunit V2	P19404	1.10
hsa:57492	ARID1B	AT-rich interaction domain 1B	Q8NFD5	1.20
hsa:60	ACTB	actin beta	P60709	1.55
hsa:6389	SDHA	succinate dehydrogenase complex flavoprotein subunit A	P31040	1.13
hsa:6390	SDHB	succinate dehydrogenase complex iron sulfur subunit B	P21912	1.47
hsa:91942	NDUFAF2	NADH:ubiquinone oxidoreductase complex assembly factor 2	Q8N183	2.97
**hsa00970 Aminoacyl-tRNA biosynthesis** (**2**)				
hsa:23438	HARS2	histidyl-tRNA synthetase 2, mitochondrial	P49590	2.07
hsa:5859	QARS1	glutaminyl-tRNA synthetase 1	P47897	1.04
**hsa03050 Proteasome** (**2**)				
hsa:143471	PSMA8	proteasome 20S subunit alpha 8	Q8TAA3	−1.13
hsa:5715	PSMD9	proteasome 26S subunit, non-ATPase 9	O00233	−2.00

## Data Availability

All data generated or analyzed during this study are included in the main text of the article and its electronic Appendix A. The mass spectrometry proteomics data were deposited to the ProteomeXchange Consortium via PRIDE partner repository [123] with dataset identifier PXD025223. NMR spectra are available on request from Eva Baranovičová.

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
