# Peer review of "Comparative Proteomic and Metabolomic Analysis of Human Osteoblasts, Differentiated from Dental Pulp Stem Cells, Hinted Crucial Signaling Pathways Promoting Osteogenesis"

_ijms, 2021, doi:10.3390/ijms22157908_

Round 1

Reviewer 1 Report

The article is focusing on osteogenic differentiation, but you work with DPSCs. How can the authors be sure that they  analyse osteoblast, but not odontoblast differentiation? Osteogenic and odontoblast differentiation are similar, but not the same.

As the authors mentioned: “Identification of dentin sialophosphoprotein (DSPP) and dentin matrix protein 1 (DMP-1) by western blotting among signaling molecules underlined the origin of DPSCs”, but these proteins are markers of odontoblast differentiation.

In that case, is it correct, for example, to expect that some signaling cascades should be regulated in the same manner as in osteogenic differentiation? “Despite the expectations, MAPK and JAK-STAT signaling cascades were apparently downregulated in osteodifferentiated cells”. But are the authors sure, that their cells are osteodifferentiated?

Many other statements, as “Selected pathways, as newly potential targets for osteodifferentiation/bone metabolism”  also looks unreasonable.

Nevertheless, the presented data is analyzed in the correct way, so I would like to recommend only a minor correction of some formulations and to add a detailed discussion of molecular mechanisms of osteogenic and odontoblast differentiation to the introduction.

The authors used “Bradford assay” for measurement of concentration of proteins, extracted with SDS. Bradford assay is not detergent compatible. Have they additionally verified that the protein concentration is equal between the samples by the protein electrophoresis?

In the label-free proteomics analysis the authors worked within mass range 50-2000 m/z. Nevertheless, it is not effective to work with m/z lower than 300 in proteomics. Maximal m/z is also might be higher (if it is possible). It is not mean that obtained data is incorrect, but I recommend  to change the m/z range for the future study – it might enhance the number of quantified proteins.

In the protein identification the authors used “Search parameters were: (i) one trypsin miscleavage”. I would like to recommend to work with at least two missed cleavages – it might also enhance number of identified proteins.

Minor remarks:

Line 177. “Ca2+ deposits..” – double dots

Author Response

Response to Reviewer #1

The article is focusing on osteogenic differentiation, but you work with DPSCs. How can the authors be sure that they  analyse osteoblast, but not odontoblast differentiation? Osteogenic and odontoblast differentiation are similar, but not the same.

As the authors mentioned: “Identification of dentin sialophosphoprotein (DSPP) and dentin matrix protein 1 (DMP-1) by western blotting among signaling molecules underlined the origin of DPSCs”, but these proteins are markers of odontoblast differentiation.

In that case, is it correct, for example, to expect that some signaling cascades should be regulated in the same manner as in osteogenic differentiation? “Despite the expectations, MAPK and JAK-STAT signaling cascades were apparently downregulated in osteodifferentiated cells”. But are the authors sure, that their cells are osteodifferentiated?

Thank you for raising this important issue. These comments are current and highly discussed.

For osteogenic induction, we used OsteoMAX-XF differentiation medium (Merck Sigma-Aldrich). Its composition, despite communication with producer, is not disclosed, yet. OsteoMAX-XF is recommended for osteodifferentiation. Based on our experiences and published manuscript, in which we compared various osteogenic differentiation media, OsteoMAX-XF™ turned out as the best based on cell morphology, mineral deposit composition and surface markers distribution (Okajčeková et al. 2020). DPSCs are mesenchymal stem cells, which could differentiate into osteoblasts, odontoblasts, or other cell types. Thus, we cannot completely rule out that at least some proportion of them differentiated into odontoblasts in vitro. We agree that osteodifferentiation is similar to odontogenic differentiation in vitro. Some authors suggested that specific successful differentiation of DPSCs with proper spatial arrangement into osteoblasts or osteoblasts is feasible only in vivo (Kim et al., 2021).

BSP II and osteocalcin, which were more abundant in differentiated cells in our study, are well-recognized bone-specific markers. However, there are ideas in the literature that it is counterproductive to discriminate bone/tooth differentiation markers because these proteins are expressed in both tissues (Kim et al., 2021). For instance, DMP-1 and DSPP (also DSP) were specified as odontogenic markers, but each of them was also identified in bones (Suzuki, 2012). In fact, these proteins are less abundant in bones than in teeth. Particularly, DSPP accumulated in teeth compared to bones (Qin, 2003). On the other hand, we reported findings from in vitro induced cell line. Perhaps future study, which would directly compare the amounts of marker proteins extracted from teeth, bone, and in vitro induced cells might clarify such uncertainty. In light of these considerations, we removed the statement “Identification of dentin sialophosphoprotein (DSPP) and dentin matrix protein 1 (DMP-1) by western blotting among signaling molecules underlined the origin of DPSCs (Fig. 6) “ from the revised text.

Analogous, signaling pathways involved in osteogenic/odontogenic differentiation overlap. For example, some authors reported that higher expression of insulin growing factor from PI3K/AKT pathway is typical for osteogenic differentiation (MacDougall and Javed, 2010), while others concluded that up-regulated PI3K/AKT pathway is accompanying odontogenesis (Zhang, 2020). Also, Wnt signaling was described in association with both osteogenesis and odontogenesis. However, supporting our line of data interpretation we detected accumulation of Wnt5A associated with osteogenesis, while Wnt10A was described as a typical odontogenic protein (Yamashiro, 2007; MacDougall and Javed, 2010). Suppression of JAK-STAT signaling during osteogenesis was already described in the literature. Therefore, we reformulated highlighted example to “MAPK and JAK-STAT signaling cascades were apparently downregulated in osteodifferentiated cells.

Herein, we would prefer to focused on in vitro differentiation of DPSCs from the osteogenic perspective.

Okajcekova, T.; Strnadel, J.; Pokusa, M.; Zahumenska, R.; Janickova, M.; Halasova, E.; Skovierova, H. A Comparative In Vitro Analysis of the Osteogenic Potential of Human Dental Pulp Stem Cells Using Various Differentiation Conditions. International Journal of Molecular Sciences 2020, 21, 2280, doi:10.3390/ijms21072280

Kim, J.; Lee, G.; Chang, W.S.; Ki, S. hyoung; Park, J.-C. Comparison and Contrast of Bone and Dentin in Genetic Disorder, Morphology and Regeneration: A Review. J Bone Metab 2021, 28, 1–10, doi:10.11005/jbm.2021.28.1.1.

Suzuki, S.; Haruyama, N.; Nishimura, F.; Kulkarni, A.B. Dentin Sialophosphoprotein and Dentin Matrix Protein-1: Two Highly Phosphorylated Proteins in Mineralized Tissues. Arch Oral Biol 2012, 57, 1165–1175, doi:10.1016/j.archoralbio.2012.03.005.

Qin C, Brunn JC, Cadena E, Ridall A, Butler WT (2003) Dentin sialoprotein in bone and dentin sialophosphoprotein gene expressed by osteoblasts. Connect Tissue Res 44(suppl 1):179–183

MacDougall, M.J.; Javed, A. Dentin and Bone: Similar Collagenous Mineralized Tissues. In Bone and Development; Bronner, F., Farach-Carson, M.C., Roach, H.I. (Trudy), Eds.; Topics in Bone Biology; Springer: London, 2010; pp. 183–200 ISBN 978-1-84882-822-3.

Yamashiro T, Zheng L, Shitaku Y, Saito M, Tsubakimoto T, Takada K, Takano-Yamomoto T, Thesleff I (2007) Wnt10a regulates dentin sialophosphoprotein mRNA expression and possibly links odontoblast differentiation and tooth morphogenesis. Differentiation 75(5): 452–462

Zhang, F.; Zhang, S.; Hu, Y.; Wang, N.; Wu, L.; Ding, M. Role of PI3K/AKT Signaling Pathway in Proliferation, Migration and Odontogenic Differentiation of Human Dental Pulp Stem Cells. Journal of Hard Tissue Biology 2020, 29, 99–104, doi:10.2485/jhtb.29.99.

Many other statements, as “Selected pathways, as newly potential targets for osteodifferentiation/bone metabolism” also looks unreasonable.

Thank you for highlighting the inaccurate statement, we revised it as: „Selected pathways potentially associated with bone metabolism “. Besides, we carefully revised the whole text of the manuscript, correcting incoherent statements.

Nevertheless, the presented data is analyzed in the correct way, so I would like to recommend only a minor correction of some formulations and to add a detailed discussion of molecular mechanisms of osteogenic and odontoblast differentiation to the introduction.

Thank you for the clear suggestion, which helped us to improve the manuscript. We added a paragraph (page 2-3, lines 87-97) dealing with the controversy of osteodifferentiation versus odontogenic differentiation to the introduction section. We also added several relatioship between DSPP and bones (page 12, line 260-263) in TGF-β cascade paragraph.

The authors used “Bradford assay” for measurement of concentration of proteins, extracted with SDS. Bradford assay is not detergent compatible. Have they additionally verified that the protein concentration is equal between the samples by the protein electrophoresis?

Thank you for spotting inaccuracy in the methods description (corrected in this revision). We used Bradford detergent compatible assay (#23246, Thermo Fisher Scientific), which tolerates up to 0.5% SDS. Such as protein samples were diluted at least10 times before the assay, SDS content was within the compatibility range, also verified with appropriate buffer blank. We added the specification of the assay in the methods section (page 20, line 552): Protein concentration was determined by the Bradford detergent compatible assay (Thermo Fisher Scientific).

Besides, we applied samples on 12% SDS-PAGE gel to control the loading, please see the figure of Coomassie-stained gel below. Furthermore, before loading peptides on the analytical column, their concentration was additionally measured to control for uneven losses during upstream sample preparation (as described in section 3.6).

In the label-free proteomics analysis the authors worked within mass range 50-2000 m/z. Nevertheless, it is not effective to work with m/z lower than 300 in proteomics. Maximal m/z is also might be higher (if it is possible). It is not mean that obtained data is incorrect, but I recommend to change the m/z range for the future study – it might enhance the number of quantified proteins.

Thank you for the valuable advice. We added important clarification in the methods section that the first quadrupole filtered precursor ions with m/z ˂ 400 (page 21, line 596): Voltage profile setting on the first quadrupole allowed efficient deflection of low mass precursor ions ˂ 400 m/z. I

ncreasing mass range above 2000 m/z on Synapt G2-Si would require switching quadrupole in a different mode with lower transmission efficiency, thus lower sensitivity.

In the protein identification the authors used “Search parameters were: (i) one trypsin miscleavage”. I would like to recommend to work with at least two missed cleavages – it might also enhance number of identified proteins.

Thank you for the suggestion. For protein digestion, we used highly efficient trypsin gold from Promega. Besides, we empirically verified on selected chromatographic runs that 2 miscleavages, in fact, yielded somewhat lower number of confident protein groups. Therefore, for our dataset, increased search space penalized more than identifying a few more peptides with 2 miscleavages.

Minor remarks:

Line 177. “Ca2+ deposits..” – double dots

Corrected.

Other corrections:

We also changed in all manuscript and supplementary tables S2 and S6 (Appendix B)........(P ≤ 0.01) and log2 transformed ratio ≥ 1.0........

to more correct version

.....(P ≤ 0.01) and at least 1-fold of log2 transformed ratio.....

Page 6, line 172

Page 24, line 692 and 698

Reviewer 2 Report

In this paper, Novakova et al. aimed to better understand the molecular mechanisms underlying osteogenesis and performed proteomic and metabolomic analysis using human dental pulp stem cell (DPSCs)-derived osteoblasts. Although data showing in this paper were somewhat limited (human dental pulp stem cell-derived osteoblasts only, single time point only), the authors carefully described their results with extensive literature search. This paper is well written, and would contribute to a number of researchers in this field.

I have one thing; there were no experimental proofs of the author’s hypothesis made by the proteomic/metabolomic analysis. In headings between 2.4, and 2.6., there were no experimental confirmation. It would be better to show experimental proofs of the author’s hypothesis. Otherwise, the contents were merely hypothesis made by the authors.

Author Response

Response to Reviewer #2

In this paper, Novakova et al. aimed to better understand the molecular mechanisms underlying osteogenesis and performed proteomic and metabolomic analysis using human dental pulp stem cell (DPSCs)-derived osteoblasts. Although data showing in this paper were somewhat limited (human dental pulp stem cell-derived osteoblasts only, single time point only), the authors carefully described their results with extensive literature search. This paper is well written, and would contribute to a number of researchers in this field.

We are grateful for the summary of essential findings and a positive evaluation of our project reported in the manuscript. We agree that comparing other differentiation fates of stem cells as well as following proteome dynamics over multiple time points would be valuable. We intend to develop these research lines in follow-up studies.

I have one thing; there were no experimental proofs of the author’s hypothesis made by the proteomic/metabolomic analysis. In headings between 2.4, and 2.6., there were no experimental confirmation. It would be better to show experimental proofs of the author’s hypothesis. Otherwise, the contents were merely hypothesis made by the authors.

Thank you for sharing the opinion. We agree that inhibition of selected enzymes would provide direct functional evidence supporting the validity of hypothesized scenario. Nevertheless, we argue that discovery proteomics/metabolomics data highlighted a list of differently abundant proteins/metabolites allowing to create plausible ideas for further research. Moreover, we verified and complemented omics results with orthogonal methods, such as western blotting and microscopy. Addressing your concerns, we modified the titles of subsections 2.4, 2.4.1, 2.4.2, and 2.5 to reflect the lack of functional proof.

Modified titles:

2.4. Selected pathways potentially associated with bone metabolism

insted of Selected pathways, as newly potential targets for osteodifferentiation/bone metabolism

2.4.1 Prevention of oxidation stress by FoxO cascade, hormone synthesis, and mineral absorption are factors that may affect the mineralization phase of osteodifferentiation

insted of Prevention of oxidation stress by FoxO cascade. Hormone synthesis and mineral absorption are factors affecting the mineralization phase of osteodifferentiation

2.4.2 Lipid metabolism and PPAR signaling are plausibly connected to energy metabolism in the context of bone development

instead of Lipid metabolism and PPAR signaling are connected to energy metabolism

2.5 Pathways potentially essential for osteodifferentiation hinted by metabolomic data

instead of Pathways incorporated to osteodifferentiation process detected using metabolomic approach

Other corrections:

We also changed in all manuscript and supplementary tables S2 and S6 (Appendix B)........(P ≤ 0.01) and log2 transformed ratio ≥ 1.0........

to more correct version

.....(P ≤ 0.01) and at least 1-fold of log2 transformed ratio.....

Page 6, line 172

Page 24, line 692 and 698

Round 2

Reviewer 2 Report

The authors replied to my questions. I mostly agreed/understood their explanation. However, as long as the authors did not show experimental proofs, this paper is skewed toward description of their hypothesis. The authors need to add careful discussion about this issue.

Author Response

Reviewer #2:

The authors replied to my questions. I mostly agreed/understood their explanation. However, as long as the authors did not show experimental proofs, this paper is skewed toward description of their hypothesis. The authors need to add careful discussion about this issue.

Thank you for appreciating our explanations. We agree that a direct functional validation of a hypothesized osteodifferentiation scenario would make our findings much more convincing. Acknowledging this limitation we seriously revised the following essential parts, which now better reflect the balance of the strength/limitation of our data:

All changes are highlighted or crossed out.

Title to: “Comparative proteomic and metabolomic analysis of human osteoblasts, differentiated from dental pulp stem cells, revealed hinted crucial signaling pathways promoting osteogenesis”.

In Abstract to: “In parallel, metabolomic data showed that aminoacyl-tRNA biosynthesis, as well as specific amino acids, likely promote osteodifferentiation.” (Page 1, line 41-43)

Conclusions to: “Understanding of molecular mechanisms which participate in various diseases and normal development is necessary both for expanding fundamental knowledge and for developing new treatment strategies. Complex proteomic and metabolomic data allowed us to identify important proteins and metabolites and to suggest their role in the osteodifferentiation, summarized in Fig. 8. In essence, we explained the molecular mechanisms behind late osteoblasts´ differentiation (mineralization phase) by discussing In our study, the molecular mechanism of a late stage of osteoblasts combined with mineralization was characterized by recruiting several known (described in the literature) as well as unconventional signaling pathways in relation to osteogenesis. Our proteomic data indicated that the osteodifferentiation in the mineralization phase is energy consuming, whereas the cells harvested energy probably from fatty acids. Furthermore, this process likely required requires intensive antioxidant protection and was plausibly is regulated by hormones. Metabolomic data enriched results by highlighting pathways related to energy metabolism and protein synthesis. Of note, the presented network of signaling pathways is largely hypothetical and should be validated in follow-up functional experiments. For instance, inhibitors of particular signaling pathways may be used for direct verification is specific molecular mechanisms indeed promote or inhibit osteodifferentiation of mesenchymal stem cells.” (Pages 22-23, lines 652-663)

Figure 8 to: Summary of the proposed molecular mechanisms of osteodifferentiation. (Page 23, line 667)